# The antiadipogenic effect of the pentacyclic triterpenoid isoarborinol is mediated by LKB1-AMPK activation

Yesenia Arcos-Reyes[1], Gildardo Rivera[2], Laurence A. Marchat[1], Juan Salas-Benito[3], Gilberto Mandujano-Lázaro[1], Moisés Monzón-Gualito[1], Esther Ramírez-Moreno[1]*

1 Instituto Politécnico Nacional, Escuela Nacional de Medicina y Homeopatía, Laboratorio de Biomedicina Molecular 2, México City, México, 2 Instituto Politécnico Nacional, Centro de Biotecnología Genómica, Laboratorio de Biotecnología Farmacéutica, Reynosa, México, 3 Instituto Politécnico Nacional, Escuela Nacional de Medicina y Homeopatía, Laboratorio de Biomedicina Molecular 3, México City, México

* maramirezmo@ipn.mx, estherramirezmoreno@yahoo.com

## Abstract

Obesity and overweight are two highly prevalent conditions worldwide, which can lead to death or produce chronic and degenerative diseases. The search for alternative therapies to control these morbidities can involve the study of metabolites obtained from plants. Particularly, pentacyclic triterpenes produce an antiadipogenic effect by affecting the expression of master regulators of adipogenesis and their signaling pathways, including LKB1-AMPK pathway. In this work, we evaluated the effect of the pentacyclic triterpene isoarborinol on adipogenesis in 3T3-L1 cells. This molecule inhibits differentiation and decreases lipid accumulation during cell differentiation in a dose-dependent manner, deregulates the expression of C/EBPβ, C/EBPδ, C/EBPα, PPARγ and SREBP-1C, causes an increase in the phosphorylation of LKB1 and AMPK, as well as a down regulation of the lipogenic factors ACC1, FAS and FABP4. These findings show that the antiadipogenic effect of isoarborinol is associated with the activation of LKB1-AMPK, leading to changes in the expression of master regulators of adipogenesis and lipogenic factors.

## Introduction

Obesity and overweight are listed by the World Health Organization (WHO) as the fourth cause of death worldwide. They are defined as an abnormal or excessive accumulation of fat that can be harmful to health. Globally, the prevalence of overweight and obesity is high and continues to increase constantly. According to the WHO, 39% of adults are overweight and 13% are obese worldwide [1]. In Mexico, obesity has been on the rise for the past 30 years; during this period, adult obesity increased by 42.2% [2]. Today, obesity is associated with the main causes of mortality, including cardiovascular diseases (20.1%), type 2 diabetes (15.2%), malignant

**Data availability statement:** All relevant data are within the paper and its Supporting Information files.

**Funding:** This work was supported by the Secretaría de Investigación y Posgrado, Instituto Politécnico Nacional (SIP-IPN)-Mexico (projects SIP-20220698 and SIP-20230869). ERM, LAM, GR, and JSB received supports from COFAA-Instituto Politécnico Nacional, EDI- Instituto Politécnico Nacional and SNII-SECIHTI. YAR received a BEIFI- Instituto Politécnico Nacional support (BEIFI A210032) and SECIHTI fellowship (CVU 1104237).

**Competing interests:** The authors have declared that no competing interests exist.

tumors (10.8%), and liver diseases (7.6%) [3]. Some strategies used to treat and prevent obesity involve exercise routines, special diets, bariatric surgery, anti-obesity medications, management of gut microbiota and psychological therapy, however several factors such as low socioeconomic status, time limitations, obesity comorbidities and the adverse effects caused by current drugs difficult treatment efficiency [4]. Therefore, it is important to search for alternative therapies to control obesity.

Metabolites obtained from plants represent an important source of new compounds against obesity. In fact, this antiadipogenic potential has been demonstrated in flavonoids as catechin, kaempferol, quercetin [5], isoflavones as genistein [6], alkaloids as berberine [7], anthraquinones as rehin [8], polyphenols as resveratrol [9], among others. Pentacyclic triterpenes are a group of compounds with multiple biological properties, and their antiadipogenic activity has already been explored. For example, it has been reported that maslinic acid [10], betulinic acid [11], oleanolic acid [12], α, β-amyrin [13], glycyrrhetinic acid [14], and ursolic acid [15,16], have an inhibitory effect on adipogenesis. Some of these compounds modify the expression of master regulators of adipogenesis, such as the transcription factors C/EBPα (CCAAT/enhancer-binding protein alpha), PPARγ (Peroxisome proliferator-activated receptor gamma) and SREBP-1C (Sterol regulatory element binding protein -1c). Their effect on adipogenesis also involve the modulation of several signaling pathways. For example, oleanolic acid exerts its effect by modulating Tyk-STAT signaling [17], while glycyrrhetinic acid seems to involve the serine/threonine kinase (AKT) pathway [14]; notably, maslinic acid [10] and α,β-amyrin [13] promote the phosphorylation of AMP-Activated Protein Kinase (AMPK), while ursolic acid acts through the LKB1-AMPK pathway [15].

The AMPK pathway plays a crucial role in regulating adipogenesis, since it participates in the sensitivity and homeostasis of lipids, glucose and insulin, as well as the control of cell cycle and cellular energy homeostasis. During adipogenesis, AMPK remains inactive due to the availability of excess nutrients and energy sources [18], while its activation by phosphorylation impacts negatively on adipogenesis. The liver kinase B1 (LKB1) is the main phosphorylator/activator of AMPK in response to increased AMP. LKB1 forms a heterotrimeric complex with two accessory subunits, Ste20-related Adapter α (STRADα) and Mouse protein 25 α (MO25α), to promote AMPK phosphorylation [19]. Some evidence indicate that AMPK activation attenuates the expression of C/EBPα, C/EBPβ and PPARγ, accompanied by a decreased expression of SREBP-1C. The phosphorylation of acetyl CoA carboxylase 1 (ACC1) and the expression of the rate-limiting enzyme carnitine palmitoyl transferase I (CPT1) are also increased. These effects are reversed by using AMPK siRNAs, confirming the inhibitory function of activated AMPK in 3T3-L1 murine adipogenesis [15]. AMPK also decreases adipogenesis via inhibition of the early mitotic clonal expansion (MCE) phase accompanied by a reduced expression of early and late adipogenic factors, including the fatty acid synthase (FAS), SREBP-1C and the adipocyte fatty acid binding protein aP2 (FABP4) [20,21].

*Petiveria alliacea L.* (Phytolacaceae) is a medicinal plant with a broad range of traditional therapeutic properties, including anti-inflammatory, antioxidant and

hypoglucemic activities [22]. We previously reported that the pentacyclic triterpene isoarborinol, a major component of *P. alliacea*, can be responsible for the antiamoebic activity of leaf extract [23], however, its antiadipogenic activity has not been documented yet. Therefore, in this work, we aim to evaluate the antiadipogenic effect of isoarborinol in 3T3-L1 adipocytes. We also study its impact on the activation of LKB1-AMPK, and the expression of the master regulators of adipogenesis C/EBPβ, C/EBPδ, C/EBPα, PPARγ, SREBP-1C and lipogenic genes ACC1, FAS, and FABP4.

Our results suggest that as other pentacyclic triterpenes, isoarborinol inhibits adipogenesis through activation of LKB1-AMPK affecting the expression of master regulators of adipogenesis and lipogenic genes. They also indicate that isoarborinol could represent a new therapeutic alternative for obesity control.

## Materials and Methods

### Isoarborinol obtention

Isoarborinol was purified from the leaves of *P. alliacea*, collected in Catemaco, Veracruz, Mexico, as previously reported [23]. Briefly, *P. alliaceae* leaves were dried at room temperature, macerated with methanol and fractionated through a silica gel column; fractions composition was analyzed by thin layer chromatography and selected fractions were used to purified isoarborinol by recrystallization method. The compound was analyzed by nuclear magnetic resonance of hydrogen and carbon ($^1$H-NMR, $^{13}$C-NMR). Finally, 0.5 mg of isoarborinol was dissolved in 2 ml DMSO, and diluted to the indicated concentrations using DMEM to evaluate its effects on 3T3-L1 cells.

### MTT assays

3T3-L1 mouse embryo fibroblasts were seeded in a 96-well plate and cultured in DMEM/high glucose supplemented with 10% *v/v* fetal bovine serum (Gibco 16000–044), 2 mM of L-Glutamine, 1X nonessential amino acids, 1 mM of sodium pyruvate, penicillin-streptomycin 1% *v/v* (10,000 unit/mL; Gibco 30–2300), in the presence of different concentrations of isoarborinol (0.36, 0.72, 1.44, 2.88, 5, and 50 μM). The cells were incubated at 37°C and 5% $CO_2$ for 24 and 48 h. In another assay, confluent 3T3-L1 cells were cultured in medium supplemented with 0.5 mM of isobutylmethylxanthine (Sigma I5879), 0.25 μM of dexamethasone (Sigma D4902), and 0.2 UI/mL of insulin (Novolin), with isoarborinol (0.36, 0.72 and 1.44 μM) to induce differentiation for 72 h at 37 °C and 5% $CO_2$. Then, the medium was removed, and cells were incubated with MTT reagent (0.40 mg/mL) for 2 h (Sigma M2128). The formazan crystals produced were dissolved with acidic isopropanol and the absorbance was measured at 570 nm, in an Epoch microplate spectrophotometer. Cells without treatment, and cells treated with 1% DMSO, were included as controls. Each experiment was performed three times by triplicate and data corresponding to control cells without treatment were taken as 100%.

### Cell differentiation

3T3-L1 cells were seeded at a density of 3x10$^4$ cells/well in 12-well plates in high glucose DMEM, containing 10% fetal bovine serum (FBS) at 37 °C and 5% $CO_2$ until 80% confluence. Adipocyte differentiation was induced (day 0) with 0.2 UI insulin (Novolin), 0.5 mM of isobutylmethylxanthine (Sigma I5879) and 0.25 μM of dexamethasone, (Sigma D4902). After 2 days of incubation, culture medium was changed to DMEM-FBS containing 0.1 UI insulin. After that, medium was replaced each 2 days with DMEM supplemented with 10% FBS. In some assays, isoarborinol was added each 2 days from day 0, at a final concentration of 0.36, 0.72 and 1.44 μM. Cells without treatment were included as controls. Each experiment was made three times by triplicate.

### Intracellular lipid determination

3T3-L1 preadipocytes (3x10$^4$/well) were seeded onto 12-well plates and induced to differentiation in the presence or absence of isoarborinol as described above. Intracellular lipids were measured through Oil Red O staining 10 days after

the induction. Briefly, cells were fixed with 3.7% formaldehyde for 30 min, washed with PBS (pH 7.4), stained with 0.12% Oil Red O solution for 15 min in darkness, and washed with distilled water. Images were captured under a Nikon inverted microscope (Eclipse TS100) after washing with distilled water. Intracellular lipids were extracted with 1 ml of isopropyl alcohol, and absorbance was read at 510 nm on the EPOCH plate reader. Each experiment was made three times by triplicate and data corresponding to control cells without treatment were taken as 100%.

## Quantification of gene expression

Cells were differentiated in 12-well plates with or without isoarborinol, as described above. Total RNA was extracted with TRIZOL on day 4 and day 10, its quantification and purity were determined in a Nanodrop. 1 µg of RNA was reverse transcribed with the High-Capacity cDNA Reverse Transcription kit (Applied Biosystems) following the manufacturer's specifications. qRT-PCR was performed with SYBR green SensiFAST™ Hi-ROX (meridian BIOSCIENCE, BIO-92005) and the StepOne, real-time PCR system (Applied Biosystem) using 1 µL cDNA (500 ng/µL) and 0.4 µL of each primer (10 µM). Primers used in this study are shown in the Table 1. All reactions were performed in triplicate and data were analyzed using the ΔΔCt method, comparing the Ct values of the gene of interest (C/EBPβ, C/EBPδ, C/EBPα, PPARγ, SREBP-1C, ACC1, FAS and FABP4) with the Ct values of the constitutive gene (β-actin). The expression of C/EBPβ, C/EBPδ was evaluated on day 4, while the expression of ACC1, FAS, FABP4, was evaluated on day 10. The expression of C/EBPα, PPARγ, SREBP-1C was evaluated on both days. Cells without treatment were used as control.

## Western blotting assays

Differentiated 3T3-L1 cells treated or not with isoarborinol were obtained at day 4 and/or day 10; they were centrifuged at 1500 rpm, at 4 ºC for 5 min and the pellet was suspended with 200 µL of cold RIPA buffer (Tris-HCl [50 mM], SDS [0.1%], NaCl [150 mM], Sodium deoxycholate [0.5%], Triton x-100 [1%]), protease inhibitors cOmplete (Merck) [2x] and PMSF [100 µM]. Samples were sonicated for 10 s at 50% ultrasonic amplitude, 5 times, with 30 s intervals, always keeping

Table 1. Genes and primers used for quantification of gene expression.

| Gene | Primer (5´- 3´) | Reference |
|---|---|---|
| C/EBPβ | Forward: GGTTTCGGGACTTGATGCA | [24] |
| | Reverse: CAACAACCCCGCAGGAAC | |
| C/EBPδ | Forward: ACTCCTGCCATGTACGACGAC | [13] |
| | Reverse: GAAGAGGTCGGCGAAGAGTTC | |
| C/EBPα | Forward: TGAAGCACAATCGATCCATCC | [25] |
| | Reverse: GCACACTGCCATTGCACAAG | |
| PPARγ | Forward: TTTTCAAGGGTGCCAGTTTC | [25] |
| | Reverse: AATCCTTGGCCCTCTGAGAT | |
| SREBP-1C | Forward: TGAAGCACAATCGATCCATCC | [26] |
| | Reverse: TAGCTGGAAGTGACGGTGGT | |
| ACC1 | Forward: ATGGGCTGCTTCTGTGACTC | [27] |
| | Reverse: CTGCAAGCCTGTCATCCTCA | |
| FAS | Forward: TTGCTGGCACTACAGAATGC | [28] |
| | Reverse: AACAGCCTCAGAGCGACAAT | |
| FABP4 | Forward: TCACCATCCGGTCAGAGAGT | [27] |
| | Reverse: CCAGCTTGTCACCATCTCGT | |
| β-actin | Forward: CCACAGCTGAGAGGGAAATC | [25] |
| | Reverse: AAGGAAGGCTGGAAAAGAGC | |

samples in an ice bath. Finally, they were centrifuged at 14,000 rpm at 4 °C for 15 min. The largest amount of supernatant was recovered without taking the pellet and stored at −20°C.

Proteins were quantitated using the BCA Protein Assay kit and an Epoch Microplate Spectrophotometer at 562 nm. Proteins (30 µg) were separated on 10% SDS-PAGE gel, and electro-transferred to a nitrocellulose membrane for 30 min. Then, the membrane was blocked with 5% skim milk (Difco) and immunoblotted for 2 h with the following anti-mouse primary antibodies, provided by Abcam: Anti-CEBP Alpha/CEBPA Rabbit mAb [EP709Y] (ab40764) (1:5000), Anti-PPAR gamma Rabbit Polyclonal antibody (ab209350) (1:5000), and by Cell Signaling Technology: Phospho-AMPKα (Thr172) (D4D6D) Rabbit mAb #50081(1:5000); AMPKα (D5A2) Rabbit mAb #5831 (1:5000); Phospho-LKB1 (Ser428) (C67A3) Rabbit mAb #3482 (1:5000); LKB1 (C60C5) Rabbit mAb #3047 (1:5000); β-Actin (13E5) Rabbit mAb #4970 (1:1000). Finally, the membrane was probed with horseradish peroxidase-labeled anti rabbit IgG secondary antibodies #7074P2 (1:10000) (Cell Signaling Technology) for 1 h. The reactive bands of target proteins were enhanced using a chemiluminescence kit (Pierce, ECL Western Blotting Substrate) and detected by the ChemiDoc MP Imaging System (BioRad). Quantitative analysis was performed by normalizing the bands of the genes of interest with the constituent gene using ImageJ® software version 1.53t (2022).

### Statistical analyses

Data of representative results from three independent experiments, performed in triplicate, were analyzed with one-way ANOVA, complemented by the Dunnett post hoc test, using GraphPad Prism® software version 8.0.2 (2019). Differences between the experimental groups were considered statistically significant at a confidence level greater than 95% ($p < 0.05$).

## Results

### Isoarborinol at 0.36, 0.72 and 1.44 µM does not affect 3T3-L1 cell viability

To determine non-toxic concentrations of isoarborinol, a cytotoxicity MTT assay was carried out probing the effect of six concentrations of isoarborinol in 3T3-L1 cultures for 24 and 48 h. Cells treated with 1% DMSO (the highest concentration of DMSO correspond to the highest concentration of isoarborinol) showed above 92% viability when compared with cells without treatment. Isoarborinol from 0.36 to 1.44 µM maintains cell survival around 80%, while the highest concentrations (2.88, 5 and 50 µM) decreased viability below 70% at both 24 and 48 h (Fig 1A, B). The effect of isoarborinol on differentiated cells viability was also evaluated using the nontoxic isoarborinol concentrations (0.36, 0.72, and 1.44 µM). For this purpose, 3T3-L1 cells were subjected to the differentiation process for 72 h in the presence of isoarborinol. As shown in Fig 1C, cell viability remained greater than 80% in all concentrations. Based on these results, we decided to evaluate the effects of 0.36, 0.72, and 1.44 µM of isoarborinol on adipogenesis.

### Isoarborinol inhibits adipogenesis in murine 3T3-L1 cells

To explore the effect of isoarborinol on adipogenesis, 3T3-L1 fibroblasts were induced to differentiation in the presence of 0.36, 0.72 and 1.44 µM of isoarborinol. On day 2, in all experimental groups, 3T3-L1 cells had fibroblast-like morphology, they are spindle-shaped, elongated, with slightly elongated nuclei (Fig 2, upper panels). This morphology was maintained until day 10 in the undifferentiated group (data not shown). Differentiated cells without treatment (Control) showed notable morphological changes from day 4, exhibiting an increase in size, rounding, and the formation of lipid droplets in the cytoplasm, which increased in number and size until the day 10, as expected for the formation of adipocytes (Fig 2, Control). Similar observations can be made in cells treated with 1% DMSO (Fig 2, DMSO 1%).

Interestingly, 3T3-L1 cells treated with increasing concentrations of isoarborinol did not undergo the typical morphological changes observed during adipogenesis of control cells, the lipid droplets were smaller and in a lower amount

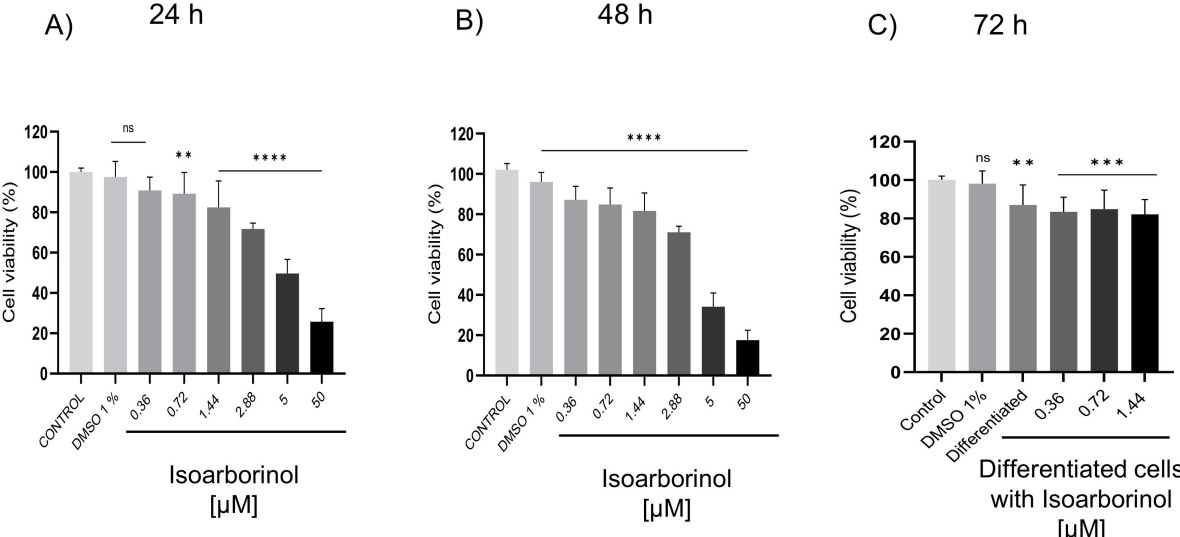

**Fig 1. Effect of isoarborinol on 3T3-L1 cell viability.** 3T3-L1 cells were treated with isoarborinol for 24 **(A)** and 48 h **(B)**, while differentiating 3T3-L1 were treated with isoarborinol for 72 h **(C)**, and cell viability was determined by MTT assay. Cells without treatment (Control) and cells treated with 1% DMSO were included. Data were analyzed using the one-way ANOVA test, and the Dunnett´s post hoc test, through the GraphPad Prism® software version 8.0.2. **** ($p < 0.0001$), ** ($p < 0.01$), * ($p < 0.05$), ns (not significant).

compared to the control groups. Interesting, the effect was concentration-dependent, with the concentration of 1.44 µM isoarborinol being the most effective (Fig 2, right panels).

### Isoarborinol decreases intracellular lipid accumulation in 3T3-L1 cells

To evaluate the effect of isoarborinol on intracellular lipid accumulation, 3T3-L1 cells treated with three different concentrations of isoarborinol (0.36, 0.72 and 1.44 µM) during the differentiation process were stained with Oil Red O on day 10. As expected, control cells have a high red staining, which denotes a great accumulation of intracellular lipids. In contrast, cells treated with isoarborinol show a lower staining as the concentration of isoarborinol increases (Fig 3A). In agreement with these observations, the quantification of intracellular Oil Red O shows that treatment with 1.44 µM isoarborinol inhibited lipid accumulation by about 30% compared to control cells, while cells treated with 0.36 µM of isoarborinol exhibited the lowest effect, with a reduction of only 16% (Fig 3B).

### C/EBP/β, C/EBPδ, C/EBPα, PPARγ, and SREBP-1C expression is down regulated in 3T3-L1 cells under the effect of isoarborinol

To better assess the effect of isoarborinol on adipogenesis, the expression of the master pro-adipogenic markers C/EBPβ and C/EBPδ, was analyzed by real-time qRT-PCR on day 4 while the expression of C/EBPα, PPARγ, and SREBP-1C, was analyzed on days 4 and 10 of differentiation (Fig 4).

On day 4, the expression of C/EBPβ was dramatically decreased by up to 60% after isoarborinol treatment, in a concentration-dependent manner, in comparison with control cells (Fig 4A), while C/EBPδexpression decreased by up 30% by effect of isoarborinol (Fig 4B). The expression of C/EBPα was also significantly downregulated after treatment with isoarborinol in a concentration-dependent manner in comparison with control cells, while the downregulation of PPARγ and SREBP-1C expression was independent of isoarborinol concentration (Fig 4C–E). On the other hand, on day 10 of differentiation, the expression of C/EBPα, PPARγand SREBP-1C were downregulated in a concentration dependent

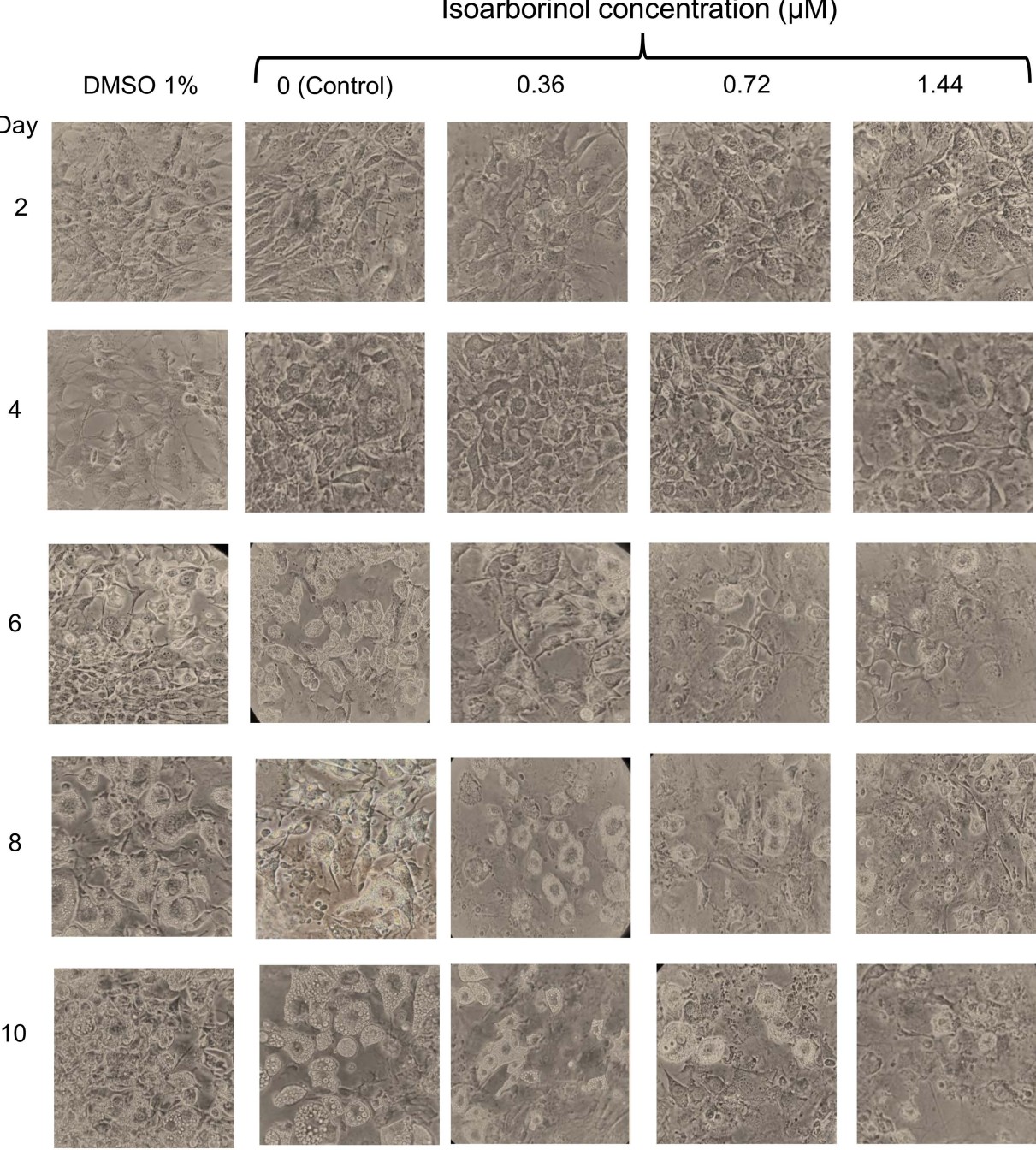

**Fig 2. Effect of isoarborinol on adipogenesis in 3T3-L1 cells.** Differentiating 3T3-L1 cells were treated with 0.36 µM, 0.72 µM and 1.44 µM isoarborinol every 48 h from day 0 to day 10 of the differentiation protocol. Differentiated cells without treatment (Control) and cells treated with 1% DMSO were included. The process was documented photographically with an inverted optical microscope at a magnification of 40X.

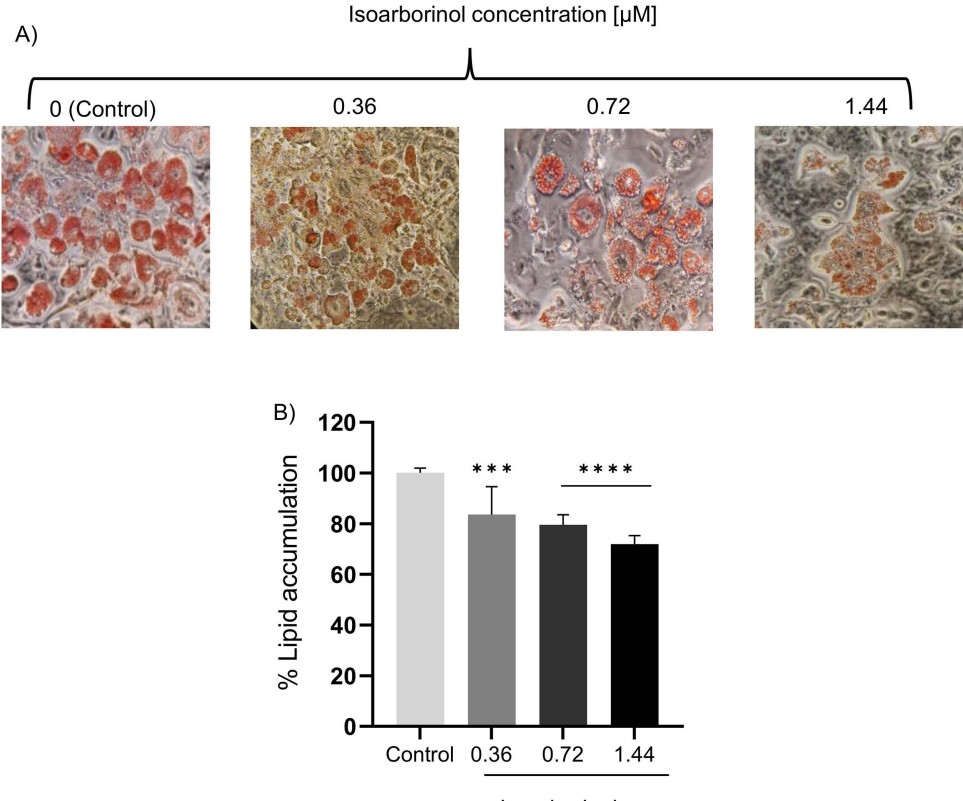

**Fig 3. Effect of isoarborinol on intracellular lipid accumulation in 3T3-L1 cells at day 10 of differentiation.** 3T3-L1 cells were induced to differentiate, in the presence or absence of isoarborinol (0.36, 0.72 and 1.44 µM). **(A)** Intracellular lipids were evidenced by Oil Red O staining. **(B)** Intracellular lipids were quantified by removing attached dye and measuring absorbance at 520 nm. Data were normalized with the control group of differentiated untreated cells taken as 100% and correspond to the mean ± standard error. Data were analyzed using the one-way ANOVA test complemented with Dunnett's post hoc test through the GraphPad prism version 8.0.2 software. **** ($p \leq 0.0001$), *** ($p \leq 0.001$).

way, with a greater effect at the highest concentration (Fig 4F–H). The same effect was observed in the protein levels for C/EBPα and PPARγ at day 10 of differentiation (Fig 5A, B).

## Isoarborinol affects the phosphorylation of AMPK and LKB1 kinases

It has been reported that pentacyclic triterpenes modulates adipogenesis and the expression of the master pro-adipogenic markers C/EBPβ, C/EBPα, PPARγ, and SREBP-1C [10,13] and through the activation of the LKB1-AMPK signaling pathway [15]. To determine whether isoarborinol exerts its antiadipogenic effect through the same molecular mechanism, changes in the phosphorylation of LKB1 and AMPK were evaluated at 4 and 10 days of the differentiation process of 3T3-L1 cells, using Western blot assays.

On day 4, cells treated with isoarborinol had a decreased ratio of phosphorylated LKB1 to total LKB1 protein, when compared to control cells. Additionally, the effect is concentration-dependent (Fig 6A, upper panel). On the other hand, a significant increase in the ratio of phosphorylated AMPK kinase to total AMPK was observed in cells treated with 1.44 µM of isoarborinol, but not with lower concentrations (Fig 6A, bottom panel).

**Day 4**

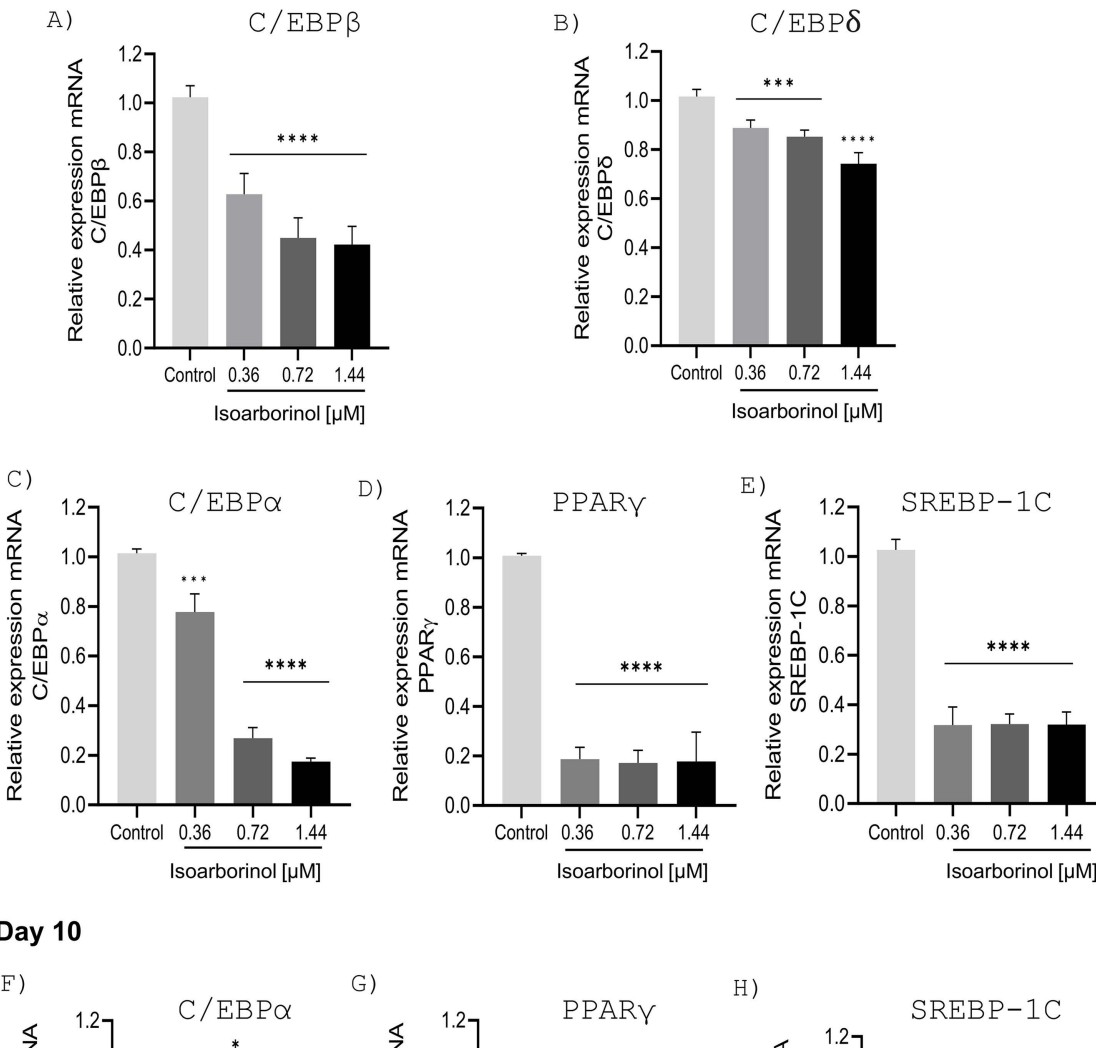

**Day 10**

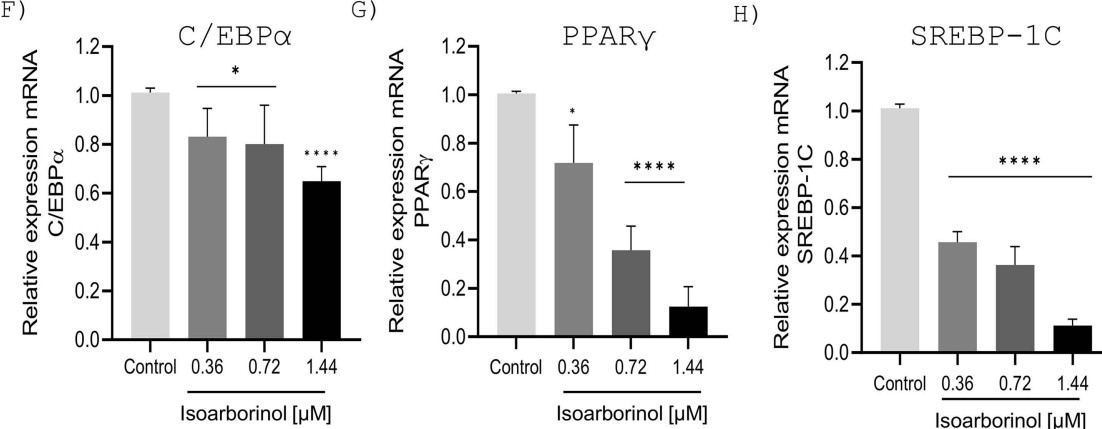

**Fig 4. Effect of isoarborinol on the gene expression levels of transcription factors in 3T3-L1 preadipocytes.** Cells were differentiated in the absence or presence of isoarborinol (0.36, 0.72 and 1.44 μM) for 4 days **(A-E)** and 10 days **(F-H)**. The graphs show the mRNA relative expression of indicate genes with respect to the concentration of isoarborinol. Each experiment was performed in triplicate. The data were analyzed using the one-way ANOVA test complemented with Dunnett's post hoc test through the GraphPad prism version 8.0.2 software, **** ($p \leq 0.0001$), *** ($p \leq 0.001$), *($p \leq 0.05$).

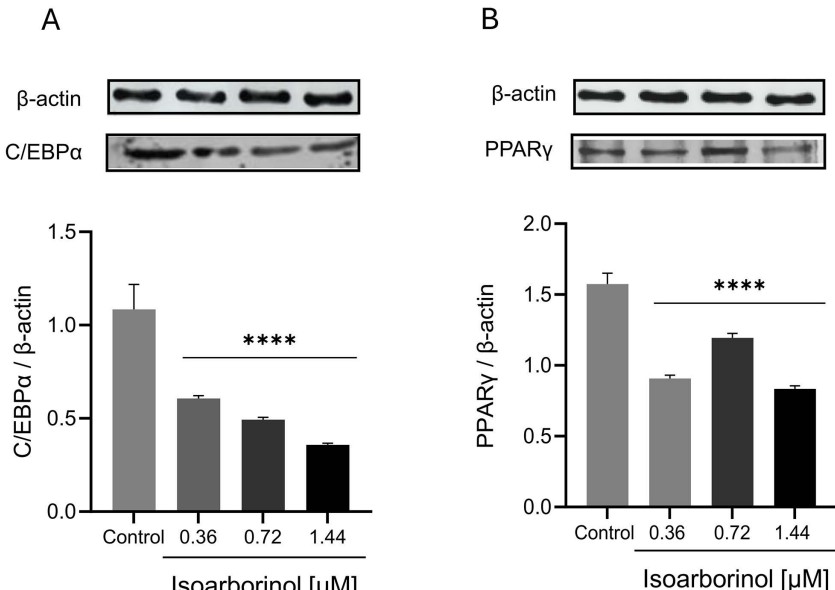

**Fig 5. Protein expression of C/EBP α and PPARγ in 3T3 cells at day 10 of differentiation.** Differentiating 3T3-L1 cells were treated with isoarborinol [0.36, 0.72 and 1.44 µM] and protein levels were determined at day 10 for western blotting assay. The graph shows the relative intensity of the expression corresponding to the proteins C/EBPα (A) and PPARγ (B) normalized with the β-actin expression; At the top of each graph the immunodetected bands are shown. Each experiment was performed in triplicate. The data were analyzed using the one-way ANOVA test complemented with Dunnett's post hoc test through the GraphPad prism version 8.0.2 software, **** p < 0.0001.

On day 10 of the differentiation process, cells under the effect of 1.44 µM isoarborinol showed an increase in the ratio of phosphorylated LKB1 to total LKB1, when compared to control cells, there was no significant effect with lower concentrations of isoarborinol (Fig 6B, upper panel). Likewise, an increase in the ratio of phosphorylated AMPK to total AMPK was observed with 0.72 and 1.44 µM isoarborinol when compared to control cells and not significant effect was produced with 0.36 µM isoarborinol (Fig 6B, bottom panel).

### Isoarborinol decreased gene expression of lipogenic proteins

Considering that isoarborinol deregulates adipogenic transcription factors and promotes the phosphorylation of AMPK and its activator LKB1, we proceeded to evaluate the gene expression of some of its downstream targets, such as ACC1, FAS, and FABP4, which are lipogenic proteins involved in *the novo* synthesis and the uptake and transport of fatty acids [29]. Isoarborinol decreases the gene expression of all three lipogenic proteins, having a concentration-dependent effect on the ACC1 protein. It produces a dramatic decrease in the expression of FAS and FABP4 at concentrations of 0.72 and 1.44 µM, while 0.36 uM of isoarborinol has little or no significant effect (Fig 7).

### Discussion

Although there are treatments against obesity, these are not very effective and they can cause serious side effects, such as kidney failure, digestive, gastric or intestinal disorders, incontinence, among others, highlighting the need for new alternative therapies for body weight control. In recent years, attention has focused on compounds of plant origin, since several secondary metabolites have antiadipogenic activity [30]. In this work we evaluated the antiadipogenic properties of the naturally occurring pentacyclic triterpene called isoarborinol, isolated from *P. alliacea* [23].

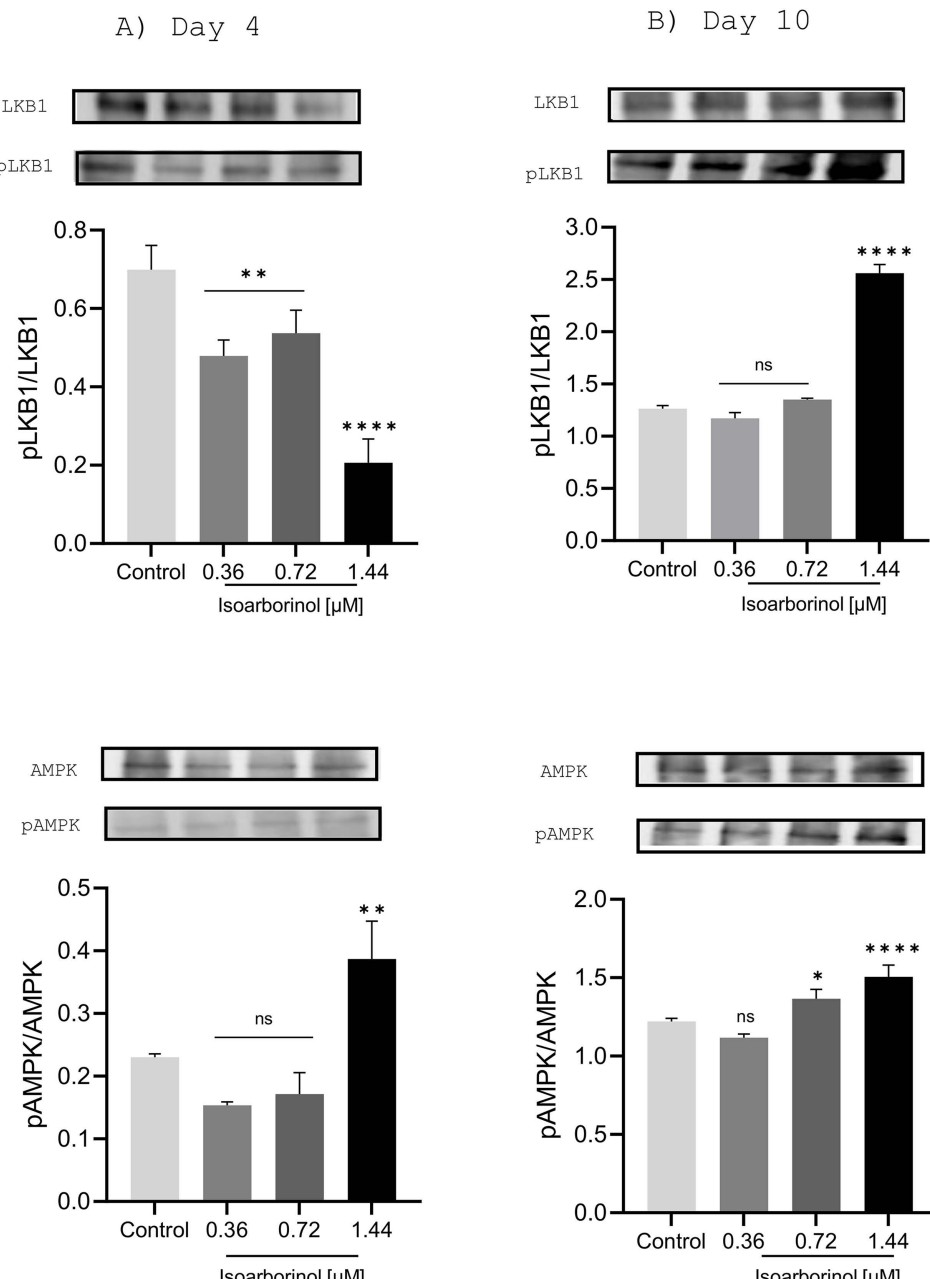

**Fig 6. Phosphorylation of LKB1 and AMPK in 3T3-L1 adipocytes treated with isoarborinol.** Differentiating 3T3-L1 cells were treated with isoarborinol (0.36, 0.72 and 1.44 µM) and protein levels of phosphorylated and total LKB1 and AMPK were determined at day 4 (A) and 10 (B) of differentiation by western blotting assay. Densitometric analysis was performed with the ImageJ® software and, pLKB1/LKB1 and pAMPK/AMPK values were determined. At the top of each graph, the immunodetected bands are shown. Each experiment was performed in triplicate. The data were analyzed using the one-way ANOVA test complemented with Dunnett's post hoc test through the GraphPad prism version 8.0.2 software, **** ($p < 0.0001$), ** ($p < 0.01$), * ($p < 0.05$), ns (not significant).

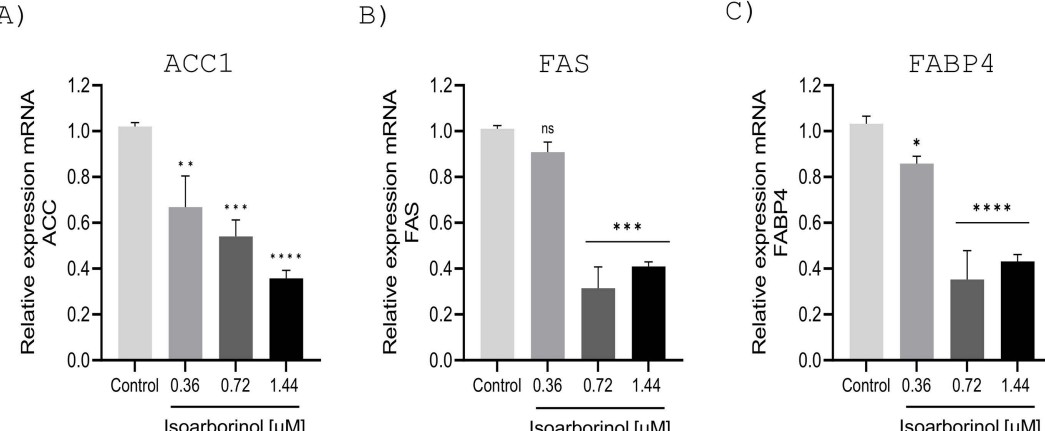

**Fig 7. Effect of isoarborinol on the gene expression levels of ACC1, FAS, and FABP4 in differentiating 3T3-L1 adipocytes.** 3T3-L1 preadipocytes were differentiated in the absence or presence of isoarborinol at concentrations of 0.36, 0.72, and 1.44 μM for 10 days; The graph shows the relationship of the relative expression of ACC1 **(A)**, FAS **(B)**, and FABP4 **(C)** genes with respect to the isoarborinol concentration. Each experiment was performed in triplicate. Data were analyzed using the one-way ANOVA test complemented with Dunnett's post hoc test through the GraphPad prism version 8.0.2 software, **** ($p \leq 0.0001$), *** ($p \leq 0.001$), *($p \leq 0.05$).

We showed for the first time that treatment with isoarborinol inhibited the morphological changes typical of 3T3-L1 adipocyte differentiation and reduced up to 30% the accumulation of intracellular lipids. Congruently, these effects were associated with a reduction of the adipogenesis regulators C/EBPβ, C/EBPδ, C/EBPα, PPARγ and SREBP-1C mRNA expression, which could correlate with a decrease in their protein expression levels, as was demonstrated for C/EBPα and PPARγ. Expression of C/EBPβ and C/EBPδ occurs at early stages of adipocyte differentiation and together they induce the expression of C/EBPα and PPARγ which are the central positive modulators of adipogenesis [31]. C/EBPα, one of the master regulators of adipogenesis, acts in concert with PPARγ to establish the phenotypes of mature adipocytes. Activation of PPARγ leads to the induction of a series of genes whose protein products mediate the cellular catabolism of triglycerides, and the uptake, transport and intracellular storage of fatty acids, adipogenesis, lipogenesis and oxidation of fatty acids, as well as glucose metabolism. Another regulator of adipogenesis is SREBP-1C, a transcription factor that participates in lipid metabolism by controlling fatty acid synthase, promoting preadipocyte differentiation, and gene expression associated with fatty acid metabolism [32].

Similar effects in adipogenesis, lipid accumulation and expression of adipogenesis regulators have been reported in 3T3-L1cells treated with other pentacyclic triterpenes, such as α, β-amyrin and ursolic acid [13,15]. However, isoarborinol produced these effects at a ≈ 10–100-fold lower concentration, since it was effective at 1.44 μM versus 10 μM for ursolic acid [15] and 125 μM (6.25 μg/ml) in the case of α, β-amyrin [13]. These differences in effective concentrations could be due to differences in experimental conditions, such as some variations on the duration of differentiation protocol and treatment frequency. Additionally, structural differences between molecules could explain the greater antiadipogenic effectiveness of isoarborinol. Unlike α,β- amyrin and ursolic acid, that have five hexanic rings (A-E), the E ring of isoarborinol is a cyclopentane, which also has an isopropyl group at C19, that is not present in α,β amyrin and ursolic acid [33]. These different groups in pentacyclic triterpenes could lead to distinct molecular interactions with target proteins and therefore distinct biological activity.

It has been reported that one of the mechanisms responsible for adipogenesis inhibition by pentacyclic triterpene involves the LKB1-AMPK pathway [13,15]. The antiadipogenic effect of isoarborinol was also associated with an increase in the phosphorylation of LKB1 and AMPK proteins. On day 4, phosphorylated LKB1 was decreased,

but the increase in phosphorylated AMPK suggests that despite low LKB1 activation, it could be enough to trigger AMPK phosphorylation in the presence of 1.44 µM of isoarborinol. Although LKB1 has been considered the primary phosphorylator of AMPK [15,34], we cannot rule out the possibility that AMPK may also be phosphorylated by other kinases. For example, calcium is known to act as a trigger for AMPK activation through the calcium/calmodulin-dependent protein kinase kinase 2 (also known as CaMKK2) [35]. Another possibility is the allosteric activation of AMPK [36]. Increased AMP and decreased ATP levels trigger AMPK activation through the direct binding of AMP to the γ subunit of AMPK, causing a conformational change that prevents phosphatases from accessing Thr172 on the α subunit and thus maintains a high level of AMPK phosphorylation [37]. On the other hand, exogenous molecules that bind to AMPK and generate conformational changes are known as direct activators. They bind to AMPK and that activate AMPK without significant changes in the ATP:AMP ratio [38]. Quercetin, AICAR, resveratrol are examples of molecules of plant origin that function as direct activators of AMPK [18]. Therefore, it is possible that isoarborinol could be acting as a direct activator of AMPK on day 4, without affecting the phosphorylation of LKB1. However, further experiments are required to verify these assumptions.

On day 10 of the differentiation process, the correlation between the increased phosphorylation of LKB1 and AMPK, suggest the participation of the LKB1-AMPK pathway in the presence of isoarborinol, and therefore inhibition of adipogenesis. This hypothesis is supported by the downregulation of some downstream molecules of the LKB1-AMPK pathway, such as ACC1, FAS, and FABP4, which are lipogenic proteins involved in the novo synthesis and the uptake and transport of fatty acids [29], which explains the significant decrease in the accumulation of intracellular lipids in adipocytes treated with isoarborinol on day 10 of differentiation.

Similarly, Vingtdeux et al. (2011) [39] reported two molecules (RSVA314 and RSVA405) that inhibit adipogenesis through the activation of AMPK accompanied by a reduced C/EBPβ expression, inhibition of C/EBPα, PPARγ and SREBP-1C, FAS and aP2 (FABP4). In another study, AMPK activation by A769662 resulted in increased phosphorylation and inactivation of ACC, reduction of lipid droplets and activation of PPARγ, C/EBPα, and early adipogenic transcription factors C/EBPβ and C/EBPδ [40].

Altogether, our results suggest that isoarborinol antiadipogenic activity could be related to the activation of the LKB1-AMPK signaling pathway. The antiadipogenic effect of other pentacyclic triterpenoids has also been associated with the LKB1-AMPK signaling pathway. He et al. [15] and de Melo et al. [13] reported that ursolic acid and α, β-amyrin increase the phosphorylation of LKB1 and AMPK, promoting the activation of the AMPK pathway and decreasing the expression of key transcription factors for adipogenesis. In the same way, Liu et al. [41] and Sung et al. [42] found that oleanolic acid activates AMPK in cancer cells, inhibiting lipogenesis, protein synthesis and aerobic glycolysis in an AMPK activation-dependent manner.

Based on our data, we propose a hypothetic working model to explain the antiadipogenic effects of isoarbrinol in 3T3-L1 murine preadipocytes. In this model, isoarborinol promotes the phosphorylation of LKB1, which phosphorylates AMPK, which in turn could inhibit and phosphorylate the transcription factors C/EBPβ and C/EBPδ, causing the reduced mRNA expression of C/EBPα, PPARγ and SREBP-1C, and reduced proteins expression of C/EBPα y and PPARγ. This could affect lipid homeostasis and the expression of the ACC1, FAS and FABP4 enzymes that are necessary for the synthesis of endogenous cholesterol, fatty acids, triacylglycerols and phospholipids, thus maintaining preadipocytes in an undifferentiated state (Fig 8). Future LKB1 and/or AMPK silencing assays would help to confirm the impact of isoarborinol on this signaling pathway during adipogenesis. Moreover, we cannot discard that the anti-adipogenic activity of isoarborinol could involve additional signaling pathways as Tyk-STAT, AKT [14,17] or other mechanisms, such as induction of apoptosis, arrest and/or delay in cell cycle progression, as it has been reported for other natural products with anti-adipogenic activity [43]. Finally, assays using primary human pre-adipocytes currently in progress will confirm the potential of isoarborinol as a new antiadipogenic treatment for obesity control.

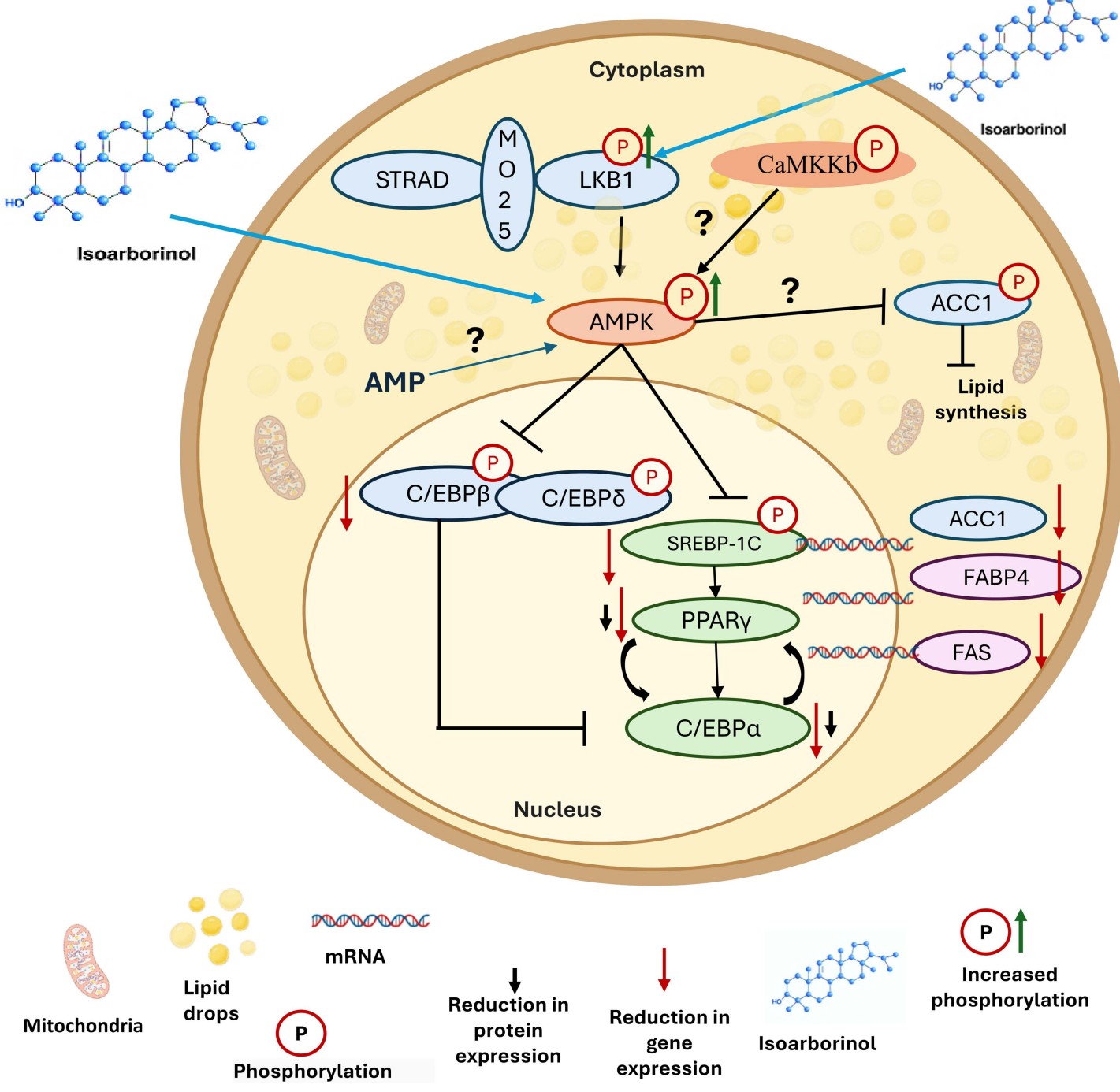

**Fig 8. Hypothetical model of the antiadipogenic effect of isoarborinol in 3T3-L1 cells.** Isoarborinol boosted phosphorylation of LKB1 and AMPK proteins, which in turn could inhibit and phosphorylate the transcription factors C/EBPβ and C/EBPδ, causing the observed reduction in mRNA expression of the transcription factors C/EBPα, PPARγ, and SREBP-1C, reduction in protein expression of C/EBPα and PPARγ, as well as reduced expression of some genes involved in adipogenic differentiation such as ACC1, FAS, and FABP4.

## Supporting information

**S1 Raw image. Raw images and data** .
(PDF)

## Acknowledgments

We thank Instituto Politécnico Nacional (IPN)-México for the scholarship (BEIFI A210032) and Secretaría de Ciencia, Humanidades, Tecnología e Innovación (SECIHTI) Mexico for the scholarship CVU 1104237 awarded to the student Yesenia Arcos Reyes. Gilberto Mandujano-Lazaro is also a postdoctoral fellowship of SECIHTI-Mexico. We also thank Dr. Cesar Augusto Sandino Reyes López and Claudia Guadalupe Benítez-Cardoza for allowing the use of their facilities.

## Author contributions

**Conceptualization:** María Esther Ramírez Moreno.

**Formal analysis:** Gildardo Rivera, Laurence A Marchat, Gilberto Mandujano-Lázaro.

**Funding acquisition:** María Esther Ramírez Moreno.

**Investigation:** Yesenia Arcos-Reyes, María Esther Ramírez Moreno.

**Methodology:** Yesenia Arcos-Reyes, Gildardo Rivera, Juan Salas-Benito, Gilberto Mandujano-Lázaro, Moisés Monzón-Gualito.

**Project administration:** María Esther Ramírez Moreno.

**Resources:** Laurence A Marchat.

**Supervision:** Laurence A Marchat, Juan Salas-Benito, Gilberto Mandujano-Lázaro.

**Writing – original draft:** María Esther Ramírez Moreno.

**Writing – review & editing:** Laurence A Marchat, María Esther Ramírez Moreno.

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
