## [Decision Letter · Decision Letter 0]

21 Jan 2025

Dear Dr. Ramírez Moreno,

Thank you for submitting your manuscript to PLOS ONE. After careful consideration, we feel that it has merit but does not fully meet PLOS ONE’s publication criteria as it currently stands. Therefore, we invite you to submit a revised version of the manuscript that addresses the points raised during the review process.

We look forward to receiving your revised manuscript.

Kind regards,

Sayed Haidar Abbas Raza

Academic Editor

PLOS ONE

Journal Requirements:

“This work was supported by the Secretaría de Investigación y Posgrado, Instituto Politécnico Nacional (SIP-IPN)-Mexico (projects SIP-20220698 and SIP-20230869). ERM, LAM, GR, and JSB received supports from COFAA-IPN, EDI-IPN and SNI-CONAHCyT. YAR received a BEIFI-IPN support (BEIFI A210032) and CONAHCYT fellowship (CVU 1104237)”

3. We note that your Data Availability Statement is currently as follows: “All relevant data are within the manuscript and in Supporting Information files.”

4. Please include captions for your Supporting Information files at the end of your manuscript, and update any in-text citations to match accordingly. Please see our Supporting Information guidelines for more information: http://journals.plos.org/plosone/s/supporting-information .

Reviewers' comments:

Reviewer's Responses to Questions

**Comments to the Author**

1. Is the manuscript technically sound, and do the data support the conclusions?

Reviewer #1: Partly

Reviewer #2: Partly

Reviewer #3: Yes

Reviewer #4: Yes

2. Has the statistical analysis been performed appropriately and rigorously?

Reviewer #1: Yes

Reviewer #2: Yes

Reviewer #3: Yes

Reviewer #4: Yes

3. Have the authors made all data underlying the findings in their manuscript fully available?

Reviewer #1: Yes

Reviewer #2: Yes

Reviewer #3: Yes

Reviewer #4: Yes

4. Is the manuscript presented in an intelligible fashion and written in standard English?

Reviewer #1: Yes

Reviewer #2: Yes

Reviewer #3: Yes

Reviewer #4: No

Reviewer #1: The manuscript is written in standard English with clarity in understanding and all data has been provided with proper statistical tests performed. While all data has been provided, it is not enough to support the claims of the study which is the reason of Major Revision.

Reviewer #2: This article showcases how isoarborinol, a pentacyclic triterpene purified from Petiveria alliacea, inhibits adipogenesis in 3T3-L1 cells by enhancing LKB1-AMPK phosphorylation and downregulating key transcription factors such as C/EBPα, PPARγ, and SREBP-1C. It elegantly demonstrates that isoarborinol significantly reduces intracellular lipid content at relatively low concentrations, suggesting potential therapeutic applications for obesity management.

The paper does not provide a thorough justification for the selected range of isoarborinol concentrations, nor does it discuss whether more granular dose-response analyses at narrower intervals could offer additional insights.

Major Flaws:

• While the authors mention that only LKB1/AMPK pathways were examined, the study lacks data on other potential mechanisms or downstream effectors (e.g., C/EBPβ, C/EBPδ, ACC, FASN). This incomplete exploration leaves open questions about the full scope of isoarborinol’s antiadipogenic pathways.

• Although beta-actin was used for normalization in Western blots, there is no clear explanation of potential loading variability or confirmation with additional housekeeping proteins. Reliance on a single internal control may compromise the accuracy of protein quantification.

• The paper limits itself to one cell model (3T3-L1) without offering any corroborative data from primary adipocytes or different adipogenic cell lines, which constrains the broader applicability and translational relevance of the findings.

• The authors do not discuss whether the reduced lipid accumulation is due primarily to suppressed adipogenesis or if isoarborinol might also induce lipolysis, apoptosis, or a cell cycle block. Without additional assays (e.g., cell cycle analysis, apoptosis markers), the mechanistic interpretation remains incomplete.

• Some concentrations and results are reported with differing significance symbols (e.g., **** vs ***), and the text does not always clarify exact p-values or confidence intervals. This inconsistency may weaken the perceived rigor of the statistical analyses.

• The discussion lacks emphasis on potential off-target effects of isoarborinol, such as interference with non-adipogenic pathways or toxicity in non-adipose tissues. No assays in other cell types or organs are described to assess broader safety or specificity.

• The paper cites previous work on other pentacyclic triterpenes but does not detail whether differences in experimental protocols (like induction cocktails, timing, or measurement endpoints) could explain variations in potency and efficacy among these related compounds.

• Although the study highlights morphological changes and lipid staining over a 10-day period, it would benefit from more detailed temporal analysis and potentially capturing earlier or later time points to see if isoarborinol’s effect is reversible or continues to intensify.

Points need to be addressed:

• Including more rigorous dose-response studies with finer increments of isoarborinol and explaining the selection of concentrations would provide a stronger foundation for the reported results.

• Investigating additional signaling molecules (such as C/EBPβ, C/EBPδ, and downstream targets like ACC or FASN) would clarify the full scope of isoarborinol’s antiadipogenic action.

• Using multiple housekeeping proteins or confirming protein loading more robustly in Western blots would enhance the reliability of the protein quantification results.

• Demonstrating isoarborinol’s effects in diverse adipogenic models or primary adipocytes, as well as exploring potential off-target activities, would strengthen the translational relevance of the findings.

• Exploring whether the reduced lipid accumulation is due to apoptosis, cell cycle arrest, or other mechanisms beyond suppressed adipogenesis (e.g., lipolysis) would offer a more comprehensive mechanistic perspective.

• Reporting p-values and confidence intervals consistently, accompanied by a clear rationale for the chosen statistical threshold, would improve the perceived rigor of the analyses.

• Testing the involvement of LKB1-AMPK more definitively, perhaps using pharmacological inhibitors or siRNA knockdowns, would robustly validate the proposed mechanism.

• Incorporating earlier or later time points and examining whether isoarborinol’s effects are reversible would enrich the understanding of the compound’s long-term or transient impact on adipogenesis.

If these concerns are sufficiently addressed, the article would be considerably strengthened, enhancing its clarity, depth, and appeal for publication in a broader scientific context.

Reviewer #3: Major Considerations:

The study needs to include the mechanistic insights into how isoarborinol specifically activates the LKB1-AMPK pathway. The authors are suggested considering experiments like using AMPK or LKB1 inhibitors, or siRNA knockdown of these proteins to support the mechanism. In addition, looking into the downstream targets of AMPK, such as ACC1 or SREBP-1c phosphorylation, could provide more information on activation of pathway.

The authors should provide more detailed information on the isolation and characterization of isoarborinol from Petiveria alliacea. This should include the extraction method, purification steps, and analytical data (e.g., NMR, mass spectrometry) to confirm the compound's identity and purity.

The authors need to investigate the phosphorylation status of LKB1 and AMPK at earlier time points (e.g., 15 minutes, 30 minutes, 1 hour, 2 hours after isoarborinol treatment) could provide insights into the immediate effects of the compound on this signaling pathway. This could help elucidate whether isoarborinol directly activates LKB1 or if there are intermediate steps involved.

The authors should consider checking the effect of isoarborinol on other key adipogenic transcription factors, such as C/EBPβ and C/EBPδ. These factors act upstream of C/EBPα and PPARγ in the adipogenic cascade, and their regulation by isoarborinol could provide a more complete picture of the compound's antiadipogenic mechanism. Experiments like gene expression analysis and protein level measurements can be done for checking.

It would be also useful investigating the effect of isoarborinol on mature adipocytes in addition to differentiating preadipocytes. Examining whether isoarborinol can induce dedifferentiation or lipid mobilization in mature adipocytes would provide a more comprehensive understanding of its potential anti-obesity effects. This could involve treating fully differentiated 3T3-L1 cells with isoarborinol and assessing changes in lipid content and adipocyte marker expression.

The authors should consider examining the effect of isoarborinol on mitochondrial function and biogenesis. AMPK activation is known to promote mitochondrial biogenesis and fatty acid oxidation.

Minor Considerations:

The authors should consider including a positive control, such as a known AMPK activator like AICAR or metformin, in their experiments. This would provide a reference point for the efficacy of isoarborinol.

The authors should discuss the potential off-target effects of isoarborinol. While the focus is on LKB1-AMPK signaling, the compound may affect other pathways that could contribute to its antiadipogenic effects.

The discussion section should include a more comprehensive comparison of isoarborinol's effects with those of other known antiadipogenic pentacyclic triterpenoids.

The authors should consider examining the effect of isoarborinol on adipokine production (e.g., leptin, adiponectin) in differentiated adipocytes.

The authors should discuss the potential limitations of using 3T3-L1 cells as a model system and consider validating key findings in primary preadipocytes or adipose tissue explants.

The conclusion section should be expanded to more clearly state the significance of the findings and potential future directions, in vivo study and the next steps for translating these results towards potential therapeutic applications.

Reviewer #4: Reyes et al studied antiadipogenic potential of an understudied pentacyclic triterpenes, isoarborinol associated with the activation of the LKB1-AMPK pathway. The study is very interesting but should address some concerns to strengthen the manuscript before publishing.

Minor:

Writing needs brushing in transition, references, results please define as to why an experiment was done, what were the condition and then the result/observation and then conclude the findings.

1st result should be a supplementary figure or can combine result1 & 2,

If possible digitally enhance the clarity of western blots.

Major:

Strongly recommend the LKB1 & AMPK silencing experiments to strengthen the association of isoarborinol to these pathway

**Do you want your identity to be public for this peer review?** For information about this choice, including consent withdrawal, please see our Privacy Policy

Reviewer #1: No

Reviewer #2: No

Reviewer #3: No

Reviewer #4: No

---

## [Author Response · Author response to Decision Letter 1]

18 May 2025

ANSWER TO REVIEWERS COMMENTS

Reviewer #1

The manuscript is written in standard English with clarity in understanding and all data has been provided with proper statistical tests performed. While all data has been provided, it is not enough to support the claims of the study which is the reason of Major Revision.

In this manuscript titled, “The pentacyclic triterpenoid isoarborinol inhibits adipogenesis through c/EBP�, PPAR� and SREBP-1C downregulation and LKB1-AMPK phosphorylation”, the authors Arcos-Reyes et al, show the safety and efficacy of isoarborinol to inhibit adipogenesis in 3T3-L1 cell system. The manuscript would be perfect for publication in PLoS One journal after addressing the following major comments:

1. Throughout the manuscript the authors compared increasing concentrations of isoarborinol with no isoarborinol treated cells. The ideal control to use in all experiments would be the vehicle i.e. DMSO since the compound was dissolved in DMSO. Care needs to be taken to make sure the increasing concentrations of isoarborinol donot increase DMSO ruling out any effects of the vehicle. In line with this, the authors would then change all control bars labeled as “0” in Isoarborinol concentration as DMSO or Vehicle for clarity of the readers.

REPLY: We thank the reviewer for his observation. We completed MTT and cell differentiation assays including 1% DMSO as control, in addition to cells without any treatment, and the results evidenced that cell viability was not affected even when using this DMSO concentration. New Figures 1 and 2.

Regarding the effect of DMSO on adipogenesis, other authors have observed that 3T3-L1 adipocyte differentiation is not compromised when DMSO concentration is under 10% (Dludla PV, Jack B, Viraragavan A, Pheiffer C, Johnson R, Louw J, Muller CJF. A dose-dependent effect of dimethyl sulfoxide on lipid content, cell viability and oxidative stress in 3T3-L1 adipocytes. Toxicol Rep. 2018 Oct 4;5:1014-1020. doi: 10.1016/j.toxrep.2018.10.002. PMID: 30364542; PMCID: PMC6197677.)

As suggested by the referee, control bars are now labeled as Control group (Untreated cells) and 1% DMSO, when cells are without treatment and treated with 1% DMSO, respectively. New figures 1 and 2.

2. While the toxicity assay is performed in 3T3-L1 fibroblasts in Figure 1, further experiments have been performed in differentiating preadipocytes. Hence, the toxicity of isoarborinol should be tested in differentiating preadipocytes. Along with the cell toxicity information, the cell proliferation effects of isoarborinol should also be checked.

REPLY: As suggested by the reviewer, we included MTT assays using differentiating adipocytes. This result shown in the new Fig. 1C, indicates that isoarborino, at the concentrations tested, is not toxic to 3T3-L1 cells in differentiation. Figure 1 C.

Regarding the effect of isoarborinol on proliferation, we consider that this is not the aim of this work. Other reports about the effects of new molecules or extracts on 3T3-L1 differentiation, only evaluate toxicity (MTT), adipogenesis and lipid accumulation (ORO staining), as we did. Additionally, it is known that cells in differentiation do not undergo cell division (Entenmann and Hauner, 1996).

Entenmann, G., & Hauner, H. (1996). Relationship between replication and differentiation in cultured human adipocyte precursor cells. The American journal of physiology, 270(4 Pt 1), C1011–C1016. https://doi.org/10.1152/ajpcell.1996.270.4.C1011

3. The authors show, through Oil Red staining, the reduction in lipogenesis. The authors can enhance this point by assaying Triglycerides in the isoarborinol treated cells. This will support their claim. Moreover, it would be interesting to check changes in lipid accumulation on treatment with isoarborinol after differentiation of cells to adipocytes. This would answer the question, does the compound inhibits adipogenesis or lipid accumulation.

REPLY: ORO staining is a classical method that is commonly used to evaluate the effect of a treatment on adipogenesis and adipocyte function, particularly lipid accumulation. Some examples include:

• Perumal, N. L., Mufida, A., Yadav, A. K., Son, D. G., Ryoo, Y. W., Kim, S. A., & Jang, B. C. (2023). Suppression of Lipid Accumulation in the Differentiation of 3T3-L1 Preadipocytes and Human Adipose Stem Cells into Adipocytes by TAK-715, a Specific Inhibitor of p38 MAPK. Life (Basel, Switzerland), 13(2), 412. https://doi.org/10.3390/life13020412

• Chae, S. I., Yi, S. A., Nam, K. H., Park, K. J., Yun, J., Kim, K. H., Lee, J., & Han, J. W. (2020). Morolic Acid 3-O-Caffeate Inhibits Adipogenesis by Regulating Epigenetic Gene Expression. Molecules (Basel, Switzerland), 25(24), 5910. https://doi.org/10.3390/molecules25245910

We consider that assaying Triglycerides would give us more information about the effects of isorborinol on lipogenic genes. The goal of our work was to study whether isoarborinol can inhibit the formation of mature adipocytes, with the aim of developing new treatment for adipose tissue expansion in obesity. Although the evaluation of the effects of isoarborinol on lipid accumulation in differentiated adipocytes is a very interesting point, this is a question we will evaluate in further research.

4. The authors show the significant reduction in key transcription factors of C/EBP, PPARgamma and SREBP-1C through quantitative PCR assay. Since RNA levels do not necessarily reflect protein levels, the RNA quantification should be accompanied by western blotting to check the protein levels in the isoarborinol treated condition. Additionally, the authors should show the variation of control in their qPCR data as the control would vary across replicates. This can be done by using the mean expression value of the controls, across replicates, as a reference to compare individual controls as well as the three isoarborinol concentration.

REPLY: We thank the reviewer for his comments. We agree with the reviewer about the fact that RNA levels do not necessarily reflect protein levels. However, many works that evaluate the effects of treatments on adipogenesis only used qRT-PCR assay to assess the expression of key regulators of adipogenesis. Some examples include:

• Lee, J. E., Cho, Y. W., Deng, C. X., & Ge, K. (2020). MLL3/MLL4-Associated PAGR1 Regulates Adipogenesis by Controlling Induction of C/EBPβ and C/EBPδ. Molecular and cellular biology, 40(17), e00209-20. https://doi.org/10.1128/MCB.00209-20

• Nobushi, Y., Wada, T., Miura, M., Onoda, R., Ishiwata, R., Oikawa, N., Shigematsu, K., Nakakita, T., Toriyama, M., Shimba, S., & Kishikawa, Y. (2024). Effects of Flavanone Derivatives on Adipocyte Differentiation and Lipid Accumulation in 3T3-L1 Cells. Life (Basel, Switzerland), 14(11), 1446. https://doi.org/10.3390/life14111446

In our work, the downregulation of C/EBPα, PPARγ and SREBP-1C gene expression agrees with the reduced adipogenesis in cells treated with isoarborinol. Therefore, we consider that western blotting would not give more relevant information. In addition, several studies on 3T3-L1 preadipocytes treated with pentacyclic triterpenes report that the mRNA expression levels of PPARγ and C/EBPα are not distinct from their protein levels. Some references are:

• Moon, M. H., Jeong, J. K., Lee, Y. J., Seol, J. W., Ahn, D. C., Kim, I. S., & Park, S. Y. (2012). 18β-Glycyrrhetinic acid inhibits adipogenic differentiation and stimulates lipolysis. Biochemical and biophysical research communications, 420(4), 805–810. https://doi.org/10.1016/j.bbrc.2012.03.078

• Sung, H. Y., Kang, S. W., Kim, J. L., Li, J., Lee, E. S., Gong, J. H., Han, S. J., & Kang, Y. H. (2010). Oleanolic acid reduces markers of differentiation in 3T3-L1 adipocytes. Nutrition research (New York, N.Y.), 30(12), 831–839. https://doi.org/10.1016/j.nutres.2010.10.001

• Choi, S. K., Park, S., Jang, S., Cho, H. H., Lee, S., You, S., Kim, S. H., & Moon, H. S. (2016). Cascade regulation of PPARγ(2) and C/EBPα signaling pathways by celastrol impairs adipocyte differentiation and stimulates lipolysis in 3T3-L1 adipocytes. Metabolism: clinical and experimental, 65(5), 646–654. https://doi.org/10.1016/j.metabol.2016.01.009

We thank the reviewer for his observation about the variation of control in our qPCR data. We followed his recommendation and made the corresponding corrections in Figure 4.

5. The authors show increased AMPK phosphorylation on isoarborinol treatment while the LKB1 shows increased phosphorylation towards day 10. The authors need to provide an explaination for the opposing LKB1 phosphorylation status at day 4 and day 10 of adipogenesis. Moreover, the phosphorylated proteins (LKB1 and AMPK) should also be compared with respective total proteins to get the extent of phosphorylation (or percentage of phosphorylation) which makes more sense than amount of phosphorylation since the protein itself can also change in amount.

REPLY: We thank the reviewer for his observation, We changed the graphs and now show the ratio of phosphorylated proteins to total protein, which allows us to better interpret the results (Figure 5).

Additionally, we also include an explanation about the opposing LKB1 phosphorylation status at day 4 and day 10 of adipogenesis and its relationship with AMPK activation. Page 18, Lines 372-389.

6. The authors claim that isoarborinol inhibits adipogenesis through LKB1-AMPK pathway but only provide the amount of proteins as evidence. To make sure LKB1-AMPK pathway is indeed involved in isoarborinol effects, their components should be reduced with isoarborinol treatment to check for rescue of the effect observed. This is crucial to show LKB1-AMPK involved in isoarborinol effects as the protein levels can change as a consequence of the treatment rather than cause the effect.

REPLY: The components of the LKB1-AMPK pathway include C/EBPβ, C/EBPδ, ACC1, FAS, FABP4. As recommended by the reviewer, we examined their expression in cells treated with isoaroborinol. The results presented in fig 4 and fig 6 showed that isoarborinol decreased gene expresssion of these transcriptional regulators and lipogenic genes, which agrees with our hypothesis that isoarborinol could inhibits adipogenesis through some mechanisms including LKB1-AMPK pathway.

Some minor comments are:

1. In the introduction paragraph 5, from lines 60-67 no reference has been provided for the claims made. Proper referencing needs to be done.

REPLY: We thank the reviewer for his observation; the corresponding references have been added in the revised version of the manuscript. Page 3, lines 60-65 in the revised version of the manuscript.

2. In figure 3 and figure 4, the y-axis of the graphs provides little information due to very few ticks provided. The graph should be corrected with increased ticks allowing the reader to infer the levels shown in the graph for each bar.

REPLY: We thank the reviewer for his observation, graphs in figures 3 and 4 have been corrected

Reviewer #2

1. This article showcases how isoarborinol, a pentacyclic triterpene purified from Petiveria alliacea, inhibits adipogenesis in 3T3-L1 cells by enhancing LKB1-AMPK phosphorylation and downregulating key transcription factors such as C/EBPα, PPARγ, and SREBP-1C. It elegantly demonstrates that isoarborinol significantly reduces intracellular lipid content at relatively low concentrations, suggesting potential therapeutic applications for obesity management.

The paper does not provide a thorough justification for the selected range of isoarborinol concentrations, nor does it discuss whether more granular dose-response analyses at narrower intervals could offer additional insights.

REPLY: We repeated the experiments on the effect of isoarborinol on the viability of 3T3-L1 cells, adding two higher concentrations of the compound. This allowed us to see that higher concentrations than those we selected significantly affect the viability of the cell culture. Figure 1

Major Flaws:

• While the authors mention that only LKB1/AMPK pathways were examined, the study lacks data on other potential mechanisms or downstream effectors (e.g., C/EBPβ, C/EBPδ, ACC1, FAS). This incomplete exploration leaves open questions about the full scope of isoarborinol’s antiadipogenic pathways.

REPLY: As recommend by the reviewer, we examined the expression of downstream effectors (ACC1, FAS, FABP4) of the LKB1/AMPK pathway, as well as upstream molecules of the adipogenesis process (C/EBPβ, C/EBPδ). The results presented in the new fig 4 and Fig 6 showed that isoarsborinol decreased gene expresssion of these transcriptional regulators and lipogenic genes.

This work is the first report about the potential of isoarborinol to affect adipogenesis. We did not evaluate other possible action mechanisms of isoarborinol, because our goal was to focus on the LKB1/AMPK pathway participation. However, we included some information regarding these possible additional mechanisms in the revised version of the discussion. Page 21, lines 422-427 in the revised version of the manuscript.

• Although beta-actin was used for normalization in Western blots, there is no clear explanation of potential loading variability or confirmation with additional housekeeping proteins. Reliance on a single internal control may compromise the accuracy of protein quantification.

REPLY: Thank you very much for your comment. We believe that adding another housekeeping protein to normalize could confirm the results. However, the results in many publications are supported by using only one. As example, we can mention this paper:

• He Y, Li Y, Zhao T, Wang Y, Sun C. Ursolic acid inhibits adipogenesis in 3T3-L1 adipocytes through LKB1/AMPK pathway. PLoS One. 2013;8(7):e70135. https://doi.org/10.1371/journal.pone.0070135 PMID: 23922935.

It is important to mention that in our assays, we quantified protein extract suing the BCA Protein Assay kit, and carefully loaded 30 µg of proteins form each condition. Moreover, actin bands are quite similar in our assays, confirming a low loading variability in the assays and we do not consider that reliance on a single internal control may compromise the accuracy of our results.

• The paper limits itself to one cell model (3T3-L1) without offering any corroborative data from primary adipocytes or different adipogenic cell lines, which constrains the broader applicability and translational relevance of the findings.

REPLY: Our manuscript is the first study about the potential of isorborinol to inhibit adipogenesis. The mouse preadipocyte cell line 3T3-L1 is very useful for exploring the potential of new antiadipogenic molecules, as other researchers have done. Some examples include:

• Moura, K. (2019). α, β-Amyrin, a pentacyclic triterpenoid from Protium heptaphyllum suppresses adipocyte differentiation accompanied by down regulation of PPARγ and C/EBPα in 3T3-L1 cells. Biomedicine & pharmacotherapy = Biomedecine & pharmacotherapie, 109, 1860–1866. https://doi.org/10.1016/j.biopha.2018.11.027

• He, Y., Li, Y., Zhao, T., Wang, Y., & Sun, C. (2013). Ursolic acid inhibits adipogenesis in 3T3-L1 adipocytes through LKB1/AMPK pathway. PloS one, 8(7), e70135. https://doi.org/10.1371/journal.pone.0070135

• Pérez-Jiménez A, Rufino-Palomares EE, Fernández-Gallego N, Ortuño-Costela MC, Reyes-Zurita FJ, Peragón J, García-Salguero L, Mokhtari K, Medina PP, Lupiáñez JA. Target molecules in 3T3-L1 adipocytes differentiation are regulated by maslinic acid, a natural triterpene from Olea europaea. Phytomedicine. 2016 Nov 15;23(12):1301-1311. doi: 10.1016/j.phymed.2016.07.001.

We agree that the effects of isoarborinol must later be confirmed in human cells and in vivo models, to determine the broader applicability and translational relevance of the findings, but this will be reported in other manuscripts. However, we included some of these ideas in the revised version of the manuscript. Page 21, lines 425-427

• The authors do not discuss whether the reduced lipid accumulation is due primarily to suppressed adipogenesis or if isoarborinol might also induce lipolysis, apoptosis, or a cell cycle block. Without additional assays (e.g., cell cycle analysis, apoptosis markers), the mechanistic interpretation remains incomplete.

REPLY: In th

---

## [Decision Letter · Decision Letter 1]

4 Jun 2025

Dear Dr. Ramírez Moreno,

Thank you for submitting your manuscript to PLOS ONE. After careful consideration, we feel that it has merit but does not fully meet PLOS ONE’s publication criteria as it currently stands. Therefore, we invite you to submit a revised version of the manuscript that addresses the points raised during the review process.

We look forward to receiving your revised manuscript.

Kind regards,

Sayed Haidar Abbas Raza

Academic Editor

PLOS ONE

Journal Requirements:

Reviewers' comments:

Reviewer's Responses to Questions

**Comments to the Author**

Reviewer #1: (No Response)

Reviewer #2: All comments have been addressed

Reviewer #3: All comments have been addressed

2. Is the manuscript technically sound, and do the data support the conclusions?

Reviewer #1: Partly

Reviewer #2: Yes

Reviewer #3: Yes

3. Has the statistical analysis been performed appropriately and rigorously?

Reviewer #1: Yes

Reviewer #2: Yes

Reviewer #3: Yes

4. Have the authors made all data underlying the findings in their manuscript fully available?

Reviewer #1: Yes

Reviewer #2: Yes

Reviewer #3: Yes

5. Is the manuscript presented in an intelligible fashion and written in standard English?

Reviewer #1: Yes

Reviewer #2: Yes

Reviewer #3: Yes

Reviewer #1: The authors have cited multiple studies where only qPCR levels of the genes such as C/EBP, PPARgamma and SREBP-1C have been provided. Here are multiple citations showing poor correlation between RNA levels and proteins (PMID: 11340206;PMID: 12952525; PMID: 27104977). Please provide western blots for the proteins.

Multiple reviewers have asked the the authors to perform treatment on differentiated cells and check lipid accumulation. Experiment needs to be performed.

Reviewer #2: (No Response)

Reviewer #3: The authors have addressed every point appropriately and also have incorporteted the figures in a way that would be beneficial to the reader.

**Do you want your identity to be public for this peer review?** For information about this choice, including consent withdrawal, please see our Privacy Policy

Reviewer #1: No

Reviewer #2: No

Reviewer #3: No

---

## [Author Response · Author response to Decision Letter 2]

22 Jul 2025

Reviewer #1: The authors have cited multiple studies where only qPCR levels of the genes such as C/EBP, PPARgamma and SREBP-1C have been provided. Here are multiple citations showing poor correlation between RNA levels and proteins (PMID: 11340206;PMID: 12952525; PMID: 27104977). Please provide western blots for the proteins. Multiple reviewers have asked the authors to perform treatment on differentiated cells and check lipid accumulation. Experiment needs to be performed.

REPLY: We thank the reviewer for his comments. We agree with the reviewer about the fact that RNA levels do not necessarily reflect protein levels.

We performed a Western blotting assay to evaluate the effect of isoarborinol on protein expression of C/EBPα and PPAR� 10 days after adipocyte differentiation. The results show that isoarborinol not only affects mRNA expression of C/EBPα and PPAR�, but also their protein expression, which supports that the anti-adipogenic effect of isoarborinol is associated with down regulation in the expression of these master regulators of adipogenesis, and associated pathways.

We have included Figure 5, which shows these results.

We have included the antibody data used in Materials and Methods, lines 177-178.

The caption for Figure 5 has been included in the Results section. Lines 285-294.

Some comments have been included in the Discussion section. Lines 356, 357, 429, and 430.

In reference to perform treatment on differentiated cells and check lipid accumulation.

REPLY: In this new version of the work, we have evaluated the changes in the expression of lipogenic genes in adipocytes treated with isoarborinol and it was found that cells treated with isoarborinol had decreased expression of ACC1, FAS and FAB1 (Fig 7), which corresponds to the decrease in intracellular lipids observed by Oil Red O staining (Fig 3). We have included this in the document and discussed these findings with those published by other authors. Page 20, lines 402-409. These experiments are consistent with our goal of evaluating the effect of isoarborinol on adipogenesis.

We understand that treating differentiated cells with isoarborinol and check lipid accumulation is a very interesting question, but this was not the goal of this study. As we explained before, this is the first study that evaluate the potential of isoarborinol for obesity treatment, and as many reports do, we decided to evaluate its effects on adipogenesis, including a possible mechanism of action. In future work, we will describe the effects of isoarborinol on differentiated adipocytes, including gene expression, lipid accumulation and molecular mechanism.

---

## [Editor Report · Decision Letter 2]

7 Aug 2025

The antiadipogenic effect of the pentacyclic triterpenoid isoarborinol is mediated by LKB1-AMPK activation

PONE-D-24-60134R2

Dear Dr. Ramírez Moreno,

We’re pleased to inform you that your manuscript has been judged scientifically suitable for publication and will be formally accepted for publication once it meets all outstanding technical requirements.

Kind regards,

Sayed Haidar Abbas Raza

Academic Editor

PLOS ONE
---

## [Editor Report · Acceptance letter]

PONE-D-24-60134R2

PLOS ONE

Dear Dr. Ramírez Moreno,

I'm pleased to inform you that your manuscript has been deemed suitable for publication in PLOS ONE. Congratulations! Your manuscript is now being handed over to our production team.

Kind regards,

on behalf of

Dr. Sayed Haidar Abbas Raza

Academic Editor

PLOS ONE